# Learning Visualization Policies of Augmented Reality for Human-Robot Collaboration

**Kishan Chandan, Jack Albertson, Shiqi Zhang**
Department of Computer Science
The State University of New York at Binghamton
{kchanda2,jalbert5,zhangs}@binghamton.edu

**Abstract:** In human-robot collaboration domains, augmented reality (AR) technologies have enabled people to visualize the state of robots. Current AR-based visualization policies are designed manually, which requires a lot of human efforts and domain knowledge. When too little information is visualized, human users find the AR interface not useful; when too much information is visualized, they find it difficult to process the visualized information. In this paper, we develop a framework, called VARIL, that enables AR agents to learn visualization policies (what to visualize, when, and how) from demonstrations. We created a Unity-based platform for simulating warehouse environments where human-robot teammates collaborate on delivery tasks. We have collected a dataset that includes demonstrations of visualizing robots' current and planned behaviors. Results from experiments with real human participants show that, compared with competitive baselines from the literature, our learned visualization strategies significantly increase the efficiency of human-robot teams, while reducing the distraction level of human users. VARIL has been demonstrated in a built-in-lab mock warehouse. VARIL is available at: https://kishanchandan.github.io/varil.html

**Keywords:** Augmented Reality, Multi-robot systems, Imitation Learning

## 1 Introduction

Robots are increasingly being used in industrial environments, such as manufacturing and warehouses, where the workspaces of people and robots are usually isolated from each other. **Human-robot collaboration (HRC)** opens up a plethora of opportunities where people and robots could complement their capabilities. Hence, robots are entering human-inhabited environments to work with humans. Humans and robots prefer different communication modalities. While humans are used to natural language and gestures in communication, robots exchange information in digital forms, such as text-based commands, resulting in a communication gap in HRC. Researchers have developed algorithms and system to bridge this communication gap using natural language [1, 2, 3, 4, 5] and vision [6, 7, 8]. **Augmented Reality (AR)** has been employed for HRC to provide an alternative communication mechanism with high bandwidth and low ambiguity [9].

AR is a technology that (1) combines real and virtual objects in a real environment; (2) runs interactively, and in real time; and (3) registers (aligns) real and virtual objects with each other [10]. Such composite views can enhance human perception of the real world, and also result in an interactive experience of the environment. Many researchers are working on uncovering the benefits of AR in HRC [11, 9]. For instance, researchers have developed AR systems that allow humans to visualize the state of the robots [12], as well as the robot intentions [13, 14]. Existing AR-based HRC systems employ a static visualization policy where the visualizations are always displayed to the user. The handcrafted static policies might suggest different visualization strategies, but those systems are not able to take important runtime factors into account (e.g., the number of robots, the current status of robots, the future intentions of robots, and the current status of human) to adapt the visualizations. The consequence is that those AR interfaces can alter people's attentional focus and result in *inattentional blindness* [15], if there is visual clutter caused by too many unwanted visualizations. Such

6th Conference on Robot Learning (CoRL 2022), Auckland, New Zealand.

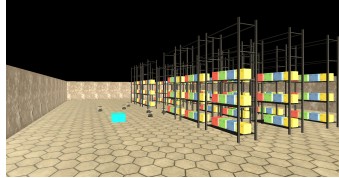 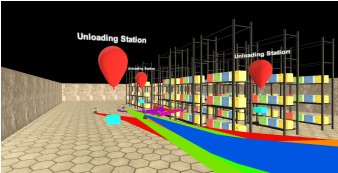 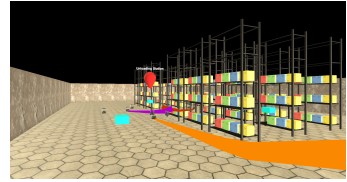

(a) No AR visualization · · · · · · (b) Full AR visualization · · · · · · (c) Learned AR visualization (ours)

Figure 1: Different AR visualization strategies in a virtual warehouse environment, where a virtual human works with a team of mobile robots on collaborative delivery tasks. (a) No AR visualization, where the human worker does not know where some robots are, and what the robots plan to do. (b) Full AR visualization, where the human worker can be overwhelmed by the visual indicators. (c) Our learned AR visualization, where the AR agent uses a learned policy to dynamically determine a visualization strategy based on the current world state of both human and robots.

concerns motivate our research on enabling our AR agent to learn a policy for dynamically selecting visualization actions given the state of the human-multi-robot system [16].

We design a framework called, *Visualizations for Augmented Reality using Imitation Learning* **(VARIL)**, for human-multi-robot collaboration in a shared environment. VARIL allows the human to track the status of a team of robots using an AR interface. In addition, VARIL learns the AR visualization policies using **Imitation Learning (IL)** [17] that dynamically selects AR visualization actions for human-multi-robot teams. To the best of our knowledge, this is the first work that employs learning to design visualization strategies of AR for human-multi-robot collaboration. In particular, we train decentralized visualization policies in a centralized fashion [18]. While training, all the states are extracted from the agents in a centralized manner, and a new policy is generated by aggregating all the states, similar to learning in a single-agent environment. The new policy learned on the aggregated dataset is then deployed to the agents in a decentralized manner, where the agents query the actions from the joint policy, by using only the current agent's state information.

We empirically evaluated our VARIL framework in a warehouse domain [19, 20], where both the human and the team of robots are engaged in a collaborative delivery task. Figure 1 shows our warehouse environment where twelve turtlebots collaborate with a human. Both the human and robots are assigned nontransferable tasks, where the robots are tasked with delivering objects to different locations (drop stations) and the human helps unload the objects from the robots waiting at drop stations. We collected a dataset of 6000 state-action pairs using the feedback provided from the "expert demonstrator". We have trained the AR agent in the simulated warehouse environment with this dataset using an iterative IL algorithm [21]. The learned AR visualization policies were compared with the static AR policy for human multi-robot teams from the literature [22], and the results suggest that the conflicts between the expert actions and the VARIL policies drastically decrease with the increase in the number of policy iterations. Additionally, we compared the total robot wait times for the intermediate policies of VARIL, and observed a significant decrease in the robot wait time between the initial and the learned AR visualization policies of VARIL.

While we observed significant improvements in the human-robot collaboration efficiency in pure simulation, we decided to evaluate how VARIL influenced the user experience. We have evaluated VARIL with 25 participants where every participant operated a virtual human in the warehouse environment. Every participant was asked to fill out a survey questionnaire that was used for subjective evaluations. Based on the participants' responses, we observed that our learned visualization strategy significantly reduced the distraction caused by extraneous visualizations. In addition to the qualitative evaluation, we also compared the task completion times during the human study. The results show significant improvement in the efficiency of human-multi-robot teams in task completion as compared to two competitive baselines from the AR-for-HRC literature [23, 22]. Finally, we present a demonstration of a human worker collaborating with three mobile robots with AR-mediated communication in a built-in-lab mock warehouse environment.

## 2   Related Work

When humans and robots work in proximity, there is a need of communication to ensure the safety and efficiency of HRC. In this section, we first describe a few communication modalities used in current human-robot systems (non-AR and AR), and then discuss existing research that applies imitation learning methods to robotics domains.

Researchers have used 2D interfaces in the past where the communication usually takes place using graphical user interfaces [24, 25, 26]. Also, natural language was used by a number of researchers for human-robot communication [5, 27]. Motion-based signaling of intent like gestures, including hand and facial have also been used for communication in human-robot teams [28, 29, 30]. Moreover, researchers studied how non-verbal social cues can improve task performance in human-robot teams [31]. Haptics, another communication modality, is largely based on physical interaction [32, 33, 34]. Even though all these modalities have their unique benefits and provide a medium for human-robot communication, the real world is 3D by nature, and these modalities are not strong in conveying spatial information, which can be very useful for HRC in shared spaces. Hence, despite those successes, AR has its unique advantages of visualizing and communicating spatial information with potentially less ambiguity and higher bandwidth [9].

AR has the capability of overlaying spatial information onto the real world and helping a human counterpart visualize the robots' internal state and intended plans for effective HRC. Researchers have developed frameworks that enable human operators to visualize the motion-level intentions of robots using AR [23, 35, 36, 37]. In another line of research, Zhu et al. augment the algorithms that control the robots to help understand the relationship between the robot's state and the environment [38]. Researchers have also developed a prototype system with a shared AR workspace that enables a shared perception [39]. In our previous work, we designed a system that enabled human users to provide feedback to robot plans visualized using AR [22]. More recently, researchers have designed different visual guidance in AR, where the humans can visualize the recommended actions from the robot (prescriptive guidance), and also visualize the state space information (descriptive guidance) to communicate why the robot recommends a particular action to the human [40]. All of the existing research employs static, predefined visualization policies. In contrast, VARIL (ours) learns a policy to select AR visualization actions (what to visualize, when, and how).

The integration of Imitation Learning (IL) with robotics has enabled the robots to learn a variety of tasks by observing expert demonstrations [41, 42]. One of the early applications of IL was to teach helicopters to fly very challenging maneuvers through providing expert demonstrations [43]. In another line of work, IL was used to enable a vehicle to mimic human driving behaviors in outdoor terrains [44, 45]. Later on, researchers used IL to learn from expert demonstrations in a simulated car driving domain [46]. A comprehensive study has been carried out to showcase the use of IL in robotics [42]. Very recently, researchers have used AR interfaces for constrained Learning from Demonstration (LfD) to allow the agents to maintain, update, and adapt learned skills [47]. They have used learning techniques to enhance the skills of the robotic systems, whereas we train our AR agent to learn an adaptive visualization policy for human-multi-robot collaboration. Similar to the discussed existing research, we use IL to help our agent learn from expert demonstrations. Instead of learning to accomplish tasks directly, our VARIL framework leverages IL to improve the efficiency of AR-based HRC, while improving user experience by avoiding visual distractions from extraneous AR visualizations.

## 3   Framework

In this section, we describe our framework called *Visualizations for Augmented Reality using Imitation Learning* (**VARIL**), and how it enables human-multi-robot teams to collaborate together in a shared environment. VARIL allows the human to track the status of a team of robots using an AR interface. Moreover, VARIL supports learning a decentralized AR visualization policy in a centralized fashion using expert demonstrations, where the AR agent uses the learned policy to dynamically select a visualization action based on the state of the human-multi-robot teams.[1] Figure 2 showcases the VARIL framework, and the interplay between the different components of VARIL.

Consider a team of robots working with a human worker in a shared space. The team of robots constantly share their state and plans using the AR interface, which is used by the human worker to track the status of the team of robots. The human worker simultaneously collaborates with the team of robots to complete the tasks. VARIL also consists of a human expert that gives demonstrations of AR visualizations at runtime, indicating what information should be visualized (or not) at specific times. Once a new policy is generated, the AR agent updates the visualizations to mimic the actions

---

[1]Our "centralized training, decentralized execution" idea was inspired by recent multiagent reinforcement learning research [18], while it is applied to an IL domain in this work.

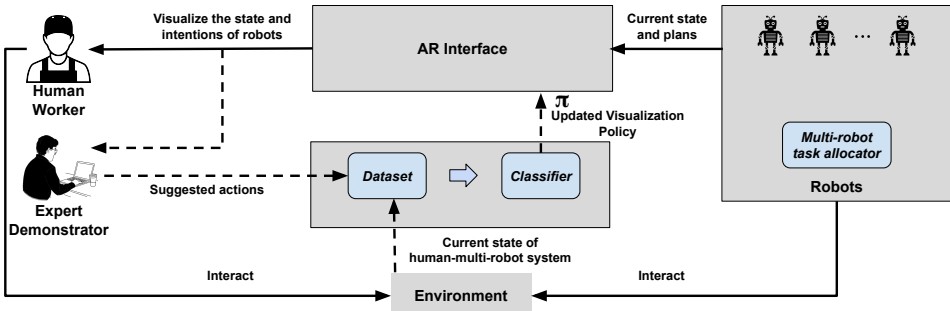

Figure 2: Overview of our VARIL framework that enables the human-multi-robot teams to collaborate in a shared environment. VARIL allows the human to track the status of a team of robots using an AR interface. Under the hood, VARIL supports learning an AR visualization policy using expert demonstrations, where the learned policy dynamically selects the actions for the AR agent.

suggested by the expert demonstrator. There are two different human entities in VARIL, one for the role of the human worker, and the other is a human expert (Figure 2). Note that the expert is involved only during the policy learning phase, and once a satisfactory policy is learned, the expert demonstrator is no longer needed.

## 3.1 VARIL Algorithm

We design a human-multi-robot collaboration system using VARIL (Algorithm 1) that enables the humans to work with a team of robots while using an AR interface to track the status of the robot teammates. The inputs to the algorithm are as follows:

- $\mathcal{P}^t$: Task pool for the robots
- $N$: Number of robots
- $\mathcal{P}^m$: Motion planner
- $\mathcal{T}^a$: Multi-robot task allocator that assigns tasks at runtime

Now, we describe in detail how our VARIL algorithm works. In Line 1, VARIL initializes $\hat{\pi}^{latest}$ to **enable all the visualizations**. Here visualizations represent all the 3D objects that are rendered in the AR interface to visualize the status of the team of robots. The intuition behind enabling all the visualizations is to enable the expert to easily identify different types of available visualizations. [2] The expert demonstrator can provide feedback on which visualizations are helpful, and which ones need to be turned on/off based on the observed state of the human-multi-robot system.

In Line 4, VARIL uses the multi-robot task allocator ($\mathcal{T}^a$) to allocate tasks ($T$) for $N$ robots, where $T_r$ is the task of the $r^{th}$ robot. Then, VARIL enters the main while-loop in Line 5 that runs until all the tasks from the task pool are completed. Inside the while-loop, VARIL enters a for-loop that runs for $N$ times, once for

---

**Algorithm 1** VARIL

**Require:** $\mathcal{P}^t, \mathcal{T}^a, \mathcal{P}^m, N, \hat{\pi}^{latest}$
1: Initialize $\hat{\pi}^{latest}$ to show all visualizations
2: Initialize empty lists: $\boldsymbol{\omega} \leftarrow \emptyset; \boldsymbol{\mathcal{I}} \leftarrow \emptyset$
3: Initialize $\mathcal{S}^{\mathcal{R}} \leftarrow \emptyset$, and $\mathcal{H}$ to $null$, where $\mathcal{S}^{\mathcal{R}}$ is the state of $N$ robots, and $\mathcal{H}$ is the state of the human worker
4: $\boldsymbol{T} = \mathcal{T}^a(\mathcal{P}^t, N)$, where $t \in \boldsymbol{T}$ is a task assigned to one robot, and $|\boldsymbol{T}| = N$ ▷ More details on $\mathcal{T}^a$ in supplementary materials.
5: **while** $\mathcal{P}^t$ is **not** empty **do** ▷ Tasks still exist in task pool
6:      **for** $i \in [0, 1, \cdots, N\text{-}1]$ **do**
7:          **if** $t_i$ is **not** complete **then**
8:              Obtain $i^{th}$ robot's configuration: $\boldsymbol{\omega}_i \leftarrow \theta(i)$
9:              $\boldsymbol{\mathcal{I}}_i \leftarrow \mathcal{P}^m(\boldsymbol{\omega}_i, t_i)$
10:              The $i^{th}$ robot follows $\boldsymbol{\mathcal{I}}_i$ using a controller
11:          **else**
12:              Update $t_i$ using $\mathcal{T}^a$
13:          **end if**
14:      **end for**
15:      $\lambda \leftarrow \mathcal{V}(\boldsymbol{\omega}, \boldsymbol{\mathcal{I}}, \hat{\pi}^{latest})$ ▷ More details on $\mathcal{V}$ in supplementary materials.
16:      $\mathcal{S}^{\mathcal{R}} \leftarrow$ obtain all robots current state
17:      $\mathcal{H} \leftarrow$ obtain current human worker state
18:      $\hat{\pi}^{latest} \leftarrow$ **PolicyUp**$(\mathcal{S}^{\mathcal{R}}, \mathcal{H})$ ▷ Update visualization policy
19: **end while**

---

each robot. At every iteration $i$ of the for-loop, VARIL first checks if the robot has completed the current task or not (Line 7). If the task is not completed, VARIL first obtains the current robot configuration ($\boldsymbol{\omega}_i$). Then, VARIL uses $\mathcal{P}^m$ to generate the intended trajectory of the $i^{th}$ robot, $\boldsymbol{\mathcal{I}}_i$. Once, the current task is complete, the robot obtains its new task using $\mathcal{T}^a$.

After the for-loop, in Line 15, the set of robot configurations, robots' intended trajectories, and the latest visualization policy are passed to the AR agent that renders the visualizations using the AR

---

[2]This is to give the human expert full observabiilty over the robots' current states and plans. If we had turned off the AR visualizations, there would be the issues, such as the human expert could not locate the robots obscured by other objects.

interface. In Lines 16 and 17, VARIL obtains the current state of all robots ($\mathcal{S}^{\mathcal{R}}$), and the current state of the human worker ($\mathcal{H}$). Finally, VARIL calls the PolicyUp function that updates the global policy $\hat{\pi}^{latest}$. This enables VARIL's AR agent to update the visualizations based on $\hat{\pi}^{latest}$ in the next iteration. In the next subsection, we explain how the PolicyUp function updates the visualization policy using IL based on the state of the human-multi-robot system and the expert demonstrations.

## 3.2 AR Policy Learning: PolicyUp

The input of Algorithm 2 includes, $\mathcal{S}^{\mathcal{R}}$, which is the state of robots; $\mathcal{H}$, the state of the human worker; $\mathcal{A}$, a set of agents for which the visualization is being learned; $\mathcal{J}$, the number of iterations for updating the policy; a dataset, $\mathcal{D}$, that stores the pairs of visited states and their corresponding actions suggested by the expert; and, $\hat{\pi}^{latest}$ that stores the latest visualization policy learned in $\mathcal{D}$.

PolicyUp enters the main for-loop in Line 1 that runs for $\mathcal{J}$ *times*. The value of $\mathcal{J}$ can be customized based on the complexity of the state space of the AR agent. In each iteration, VARIL enters another for-loop in Line 2 that runs once for each agent, $\alpha$, where $\alpha \in \mathcal{A}$. Inside the inner for-loop, PolicyUp first gets a policy $\pi_j$ based on a decision rule, which decides if the expert policy or the novice policy

---

**Algorithm 2** PolicyUp($\mathcal{S}^{\mathcal{R}}, \mathcal{H}$)

**Require:** $\mathcal{J}, \mathcal{A}, \mathcal{D}, \hat{\pi}^{latest}$
1: **for** $j = 0$ to $\mathcal{J}$ - 1 **do**
2:     **for** each agent $\alpha$ in $\mathcal{A}$ **do**
3:         $\hat{\pi}_j \leftarrow \hat{\pi}^{latest}$ ▷ Latest visualization policy is used in the current iteration.
4:         Let $\pi_j = \beta_j \pi^* + (1 - \beta_j)\hat{\pi}_j$
5:         Sample $T$-step trajectories using $\pi_j$
6:         Obtain expert feedback $f$ on the observed states
7:         Update visualization for agent $\alpha$ based on $\pi_j$
8:         Get dataset $\mathcal{D}_j^\alpha$ = state of human-multi-robot system ($\mathcal{S}^{\mathcal{R}}, \mathcal{H}$), and expert suggested feedback (actions)
9:     **end for**
10:     Aggregate datasets: $\mathcal{D} \leftarrow \mathcal{D} \bigcup \mathcal{D}_j^\mathcal{A}$ ▷ Combine individual datasets from all agents as one dataset
11:     Train classifier $\hat{\pi}_{j+1}$ on $\mathcal{D}$ ▷ Common updated policy generated for all agents
12:     Update $\hat{\pi}^{latest}$ with $\hat{\pi}_{j+1}$ ▷ Global policy is updated for visualization
13: **end for**
14: **return** $\hat{\pi}^{latest}$

---

will be used for the current iteration. The decision rule uses the value of $\beta$ that decays over the iterations, and at the beginning, the value of $\beta = 1$, and hence expert policy actions are preferred over the novice policy actions. As the value of $j$ becomes closer to $\mathcal{J}$, the value of $\beta$ becomes closer to $0$. Intuitively this makes sense because at the beginning the novice policy is not mature enough and can lead to many mistakes, and hence the expert policy is used at the beginning. Then using the policy $\pi_j$, VARIL samples $T$-step trajectories in Line 5. During this phase, the visualizations are constantly updated for every agent $\alpha$ based on the current policy $\pi_j$ (Line 7). Finally, for every agent, VARIL stores the visited states as well as the suggested actions by the expert in $\mathcal{D}_j^\alpha$, where $j$ represents the $j^{th}$ iteration of the algorithm.

After completion of one iteration of the inner for-loop, the algorithm reaches Line 10 for data aggregation. Here, VARIL combines all the datasets stored in the current iteration ($\mathcal{D}_j^\mathcal{A}$) with the dataset from the previous iteration ($\mathcal{D}$). Note that even though the datasets were obtained from different agents, the data aggregation step considers them as they were obtained from a single agent. Such abstraction of data allows the algorithm to generate a comprehensive dataset based on the experiences of all the agents, and also enables VARIL to **train the visualization policies in a centralized manner with fully decentralized execution**. Once the dataset is aggregated, VARIL trains a classifier $\hat{\pi}_{j+1}$ on $\mathcal{D}$ in Line 11. The global policy $\hat{\pi}^{latest}$ is updated with the newly generated policy, which is then used by all the agents in the next iteration of the algorithm.

The next section provides details on the experiments conducted to evaluate VARIL in the human-multi-robot collaboration scenario.

## 4 Experiments

We conducted experiments in a simulated warehouse environment that we developed using Unity. With the experiments, we aim to evaluate **two hypotheses**: I) VARIL improves the overall efficiency in human-multi-robot team task completion, determined by the slowest agent of a team, in comparison to other AR-based methods from the literature that employ static visualization policies; and II) VARIL provides better user experience for humans in human-multi-robot collaboration tasks compared to competitive AR-based methods from the literature. We have evaluated both Hypothesis-I and Hypothesis-II with human participants collaborating with multiple robots in the simulated warehouse environment.

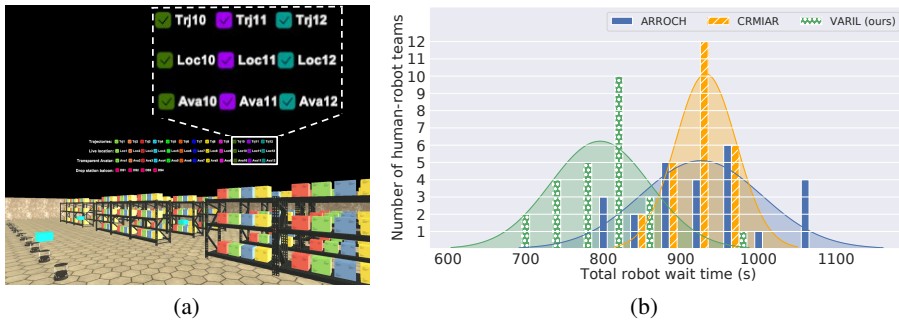

|           |           |
|-----------|-----------|
| (a)       | (b)       |

Figure 3: (a) Interface used by the expert to give demonstrations. (b) Comparison of VARIL (ours) with two baselines to show the distribution of total waiting time (second) of all robots.

**Participants:** Twenty-five participants of ages 20-30, participated in the experiment, including six females and nineteen males. Every participant was given a $10 gift card for participating in the experiments. Each participant conducted three trials using three methods, including VARIL, that were randomly ordered for each participant. Before starting the actual trials, each participant completed a short dummy trial that was scaled down to a mini warehouse that includes only one robot. The role of the dummy trial was to acquaint the participants with the environment and to teach the participant to control the virtual human. The experiments were approved by the Institutional Review Board (IRB) of our institution.

## 4.1 Experiment Setup

We deployed the simulated warehouse environment to a web server to facilitate the online experiment using the WebGL build of the Unity project. In our experiments, the human participants were able to teleoperate the virtual human around the environment through the input of the *W, A, S,* and *D keys* along with using the mouse to rotate the virtual human's field of view. Each robot is assigned three boxes for each trial and once the last box has been dropped off by the robot, it will begin navigating to its starting position. The trial is completed once all twelve robots have delivered their three boxes and have reached their starting position.

**Data Collection and Policy Learning:** We collected a dataset of 6000 demonstrations provided by a human expert at runtime during which a human worker was working with a team of robots to collaborate on a delivery task. In our case, the human expert is the first author of this paper, who knows this domain well, and we designed a virtual human to perform the task of the human worker. The human worker and the human expert both were provided with an AR interface to track the robots. The human worker used the AR interface to collaborate with the robots, whereas the role of the human expert was to provide feedback on the visualizations seen using the AR interface.

The human expert was provided with a simple interface as shown in Figure 3(a) to input the feedback (which visualizations the human expert thinks should be to enabled or disabled) on the visualizations for the team of robots, as well as the drop station balloons. The interface provides a checkbox for each robot's trajectory, live location, and transparent avatar as well as a checkbox for each drop station balloon. The expert feedback is obtained in real time and the environment is not paused. The expert does not have access to the internal state representation, and the feedback is completely based on the human expert's observations on the environment. Along with the state of the human worker and the team of robots, the feedback (actions) of the human was combined to collect the dataset. A snapshot of the state-action pairs was extracted every four seconds, and after every 25 state-action pairs, the dataset was aggregated with the dataset from the previous iteration. Using VARIL, we ran a total of two hundred forty iterations, creating the entire dataset of the state-action pairs. At every iteration, we train the AR agent with a multi-class SVM [48]. More details on the **state-action** space of the AR agent are provided in the supplementary materials.

**Baselines:** For baselines, we have replicated the visualizations of two different systems from the literature, called ARROCH [22], and CRMIAR [23] [3] to make comparisons of their visualization

---

[3] We create the acronym of CRMIAR that indicates "Communicating Robot Motion Intent with Augmented Reality" for the simplicity of referring to the method.

strategies with the learned visualization from VARIL. ARROCH was designed for human-multi-robot systems, where a human can use an AR device to track the status of the robots. Our implementation of the ARROCH system has the following visualizations: the trajectories to show the motion intentions, robot avatar to show the live location, and a transparent robot avatar to show the direction of the motion of the robot. On the other hand, CRMIAR was a single robot visualization system, but we have scaled it to enable tracking the team of robots in our warehouse environment. In our implementation of the CRMIAR system, where we have combined the NavPoints, and Utilities designs of CRMIAR to provide the following visualizations: the trajectories of robots to show their planned motion intentions, a map to show the relative location of the robots with respect to the human, and arrows to point to the robots that are not in the field of view. We use both ARROCH and CRMIAR in our experiments to evaluate the efficacy of VARIL. In the supplementary materials, we provide the figures of the both ARROCH and CRMIAR baseline implementations.

## 4.2 Real-world Demonstration

For demonstration, a built-in-lab mock warehouse was constructed in a 2500 square foot room. Three turtlebot robots were used to mimic the human-multi-robot collaborative delivery task similar to the Unity environment. The speed of the Turtlebots were set to 0.5 m/s. The human worker was provided with an AR device (tablet) with a screen size of 10 inches. Figure 4 shows the "warehouse" environment for the demonstration of VARIL, where a human worker is holding an AR device to track the status of three turtlebots. A demonstration video is available in the project webpage – see the abstract.

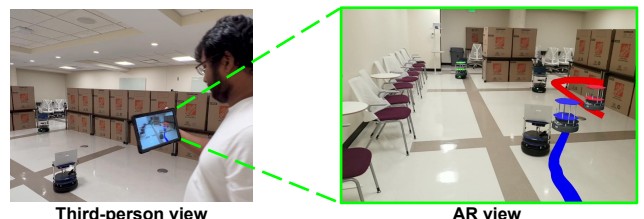

**Third-person view**      **AR view**

Figure 4: Mock warehouse and AR visualizations that include robot avatars and trajectories in color.

## 4.3 Results

**Objective Results:** Figure 3(b) shows the histogram, where the x-axis represents the total wait time of all the robots in seconds, and the y-axis represents the number of human-robot teams with the corresponding robot wait times. From the figure, it can be observed that for most human-robot teams, VARIL had the shortest robot wait times as compared to ARROCH and CRMIAR. Also, we plotted the Gaussian curves to show the distribution of the data points for all the methods. We also analyzed the statistical significance, in every trial, we sum up the task completion time of all the robots. We found that VARIL performed **significantly better** than both ARROCH and CRMIAR, where $0.01 < p-value < 0.05$. Additionally, by comparing the total robot wait times, we observed that the wait times for robots in VARIL were significantly shorter than the baselines. All of these results support Hypothesis-I that states VARIL improves the human-robot team's task completion efficiency.

**Subjective Results:** At the end of each trial, the participants were asked to fill out a survey form indicating their qualitative opinion over the six different questions to validate Hypothesis-II. The figure and the descriptive subjective analysis are provided in the supplementary materials. We observed significant improvements from the study supporting Hypothesis-II on user experience, that the visualizations in VARIL provided a better user experience. Collectively, the results convey that the VARIL visualizations help the participant better keep track of the robot status, proved useful towards task completion, and were less distracting, while the participants enjoyed using the system.

**Policy Evaluation:** VARIL is the first work on Imitation Learning-based visualization policy learning for AR, and hence we do not have learning baselines to make comparisons for the generated policy. To overcome this shortcoming, we have evaluated our learned policy to find the discrepancy between the suggested expert actions as well as the actions suggested by both the ARROCH static visualization policy, and the intermediate policies of VARIL. Figure 5(a) shows the number of policy iterations on the x-axis, and the number of discrepancies on the y-axis. The discrepancy is a measure of the deviation of the visualization policies from the expert's policy. From the figure, we can observe that the early policies of VARIL had more discrepancy, and this is because at the

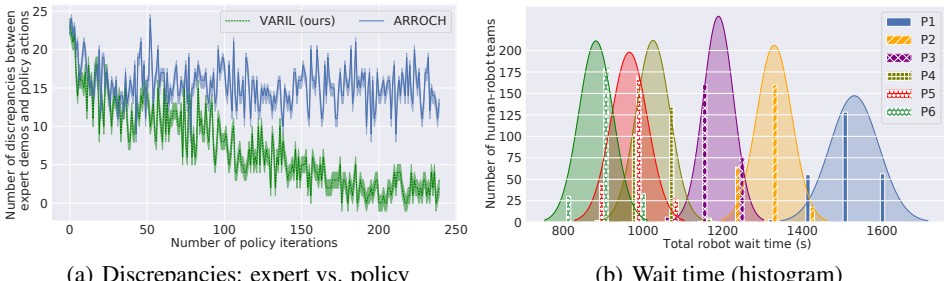

(a) Discrepancies: expert vs. policy      (b) Wait time (histogram)

Figure 5: (a) The number of discrepancies between the expert demonstrator's suggested actions with both the visualization policies of ARROCH (baseline) and VARIL with respect to the number of iterations. Each instance of a human expert disabling a visualization is considered a **discrepancy**; (b) A histogram to show the distribution of total waiting time (second) of all robots in simulation with respect to six intermediate policies of VARIL.

beginning the policy's poor quality makes it less effective in selecting visualization actions. With more interaction with the expert, the policy becomes mature, and tries to mimic the actions suggested by the expert. It can be observed that the discrepancies drastically decrease with the increase in the number of policy iterations. On the other hand, the increase of policy iteration has no effect on the number of discrepancies between ARROCH and expert actions, because ARROCH employs a static visualization policy. The final policy was used in the experiments to evaluate the learned visualization policy against the static policies from the baselines.

Furthermore, we evaluated Hypothesis-I in simulation to investigate how the intermediate policies of VARIL affect the human-robot team's task completion time. Figure 5(b) shows the histogram of the total robot wait times, for six different policies (P1 - P6), where every policy was extracted after 50 policy iterations. A total of 250 trials were run for every policy, and we observed improvements in the robot wait times denoting the improvement of efficiency in human-robot teams.

## 5 Conclusion

In this paper, we present our framework, VARIL (*Visualizations for Augmented Reality using Imitation Learning*), that for the first time introduces a learning-based Augmented Reality (AR) visualization strategy for human-multi-robot collaboration. We have designed a system that learns a decentralized visualization policy using imitation learning in a centralized manner, where the AR agent dynamically selects a visualization action based on the state of the human-multi-robot system. We compared our framework with competitive baselines from the literature, and the results suggest that VARIL significantly increases the efficiency of human-robot collaboration, and reduces the distractions caused by extraneous information in AR.

**Limitations:** While we have demonstrated VARIL using real robots, building and evaluating such AR-for-HRC systems in the real world can be difficult. For example, the human worker needs to walk between drop stations, where each navigation action might take a long time. Adding to this complexity, the robots need to collaborate in real time to complete the tasks where a single robot getting stuck to a dynamic obstacle can terminate the trial prematurely. To avoid such issues, we designed an open source simulator that can simulate a warehouse environment including a team of robots. Even though such a simulator does not overcome all the challenges for human-robot collaboration, we expect that this will enable AR-for-HRC researchers to focus on the crucial aspects of algorithm design rather than the tedious software development work. In the future, we aim to evaluate the user-experience performance of VARIL through questionnaires and explore VR-based human-robot collaboration [49], while leveraging contextual scene information [50].

## Acknowledgement

This work has taken place in the Autonomous Intelligent Robotics (AIR) group at The State University of New York at Binghamton. AIR research is supported in part by the National Science Foundation (NRI-1925044), Ford Motor Company, OPPO, and SUNY Research Foundation. The work of Albertson was supported in part by NSF REU (NRI-2032626).

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
