# OpenReview forum: "Learning Visualization Policies of Augmented Reality for Human-Robot Collaboration"
_robot-learning.org/CoRL/2022/Conference — CoRL 2022 Poster_

### Official Review · Reviewer_TGsr · 2022-07-22

**Originality:** Good
**Technical Quality:** Good
**Clarity Of Presentation:** Fair
**Impact:** 2

**Recommendation:**

Weak Accept: I recommend accepting the paper, but will not argue for my recommendation if the majority of other reviewers have a different opinion.

**Summary:**

This paper presents VARIL, an imitation learning-based Augmented Reality (AR) visualization strategy for human-multirobot collaboration. Main idea is to learn a visualization policy from an expert user, so that the visualized information in AR is adequate to perform the collaborative task with multi robots. The resulting system is a framework, where a visualization policy is learned and deployed from human demonstrations. With a user study that invovled 25 participants, the simulation results show that VARIL can increase the efficiency of human-robot teams in delivery tasks within warehouse scenarios, while reducing the mental effort required by the human users.

**Issues:**

Address the comments described above.

**Quality Of The Limitations Section:**

Limitations are not well addressed

**Reviewer Expertise:**

4: The reviewer is confident but not absolutely certain that the evaluation is correct

**Robotics Focus:**

Sufficient demonstration on hardware

**Strengths And Weaknesses:**

Strengths:

- The paper presents the first system to learn visualization policy from human demonstrations.
- Experimental results in simulation show that VARIL can reduce the mental workload of the user and be more efficient, when compared to existing approaches for visualization in AR.
- There are plans to open source the warehouse simulation environment.

Weakness:

- The paper lacks description about challenges, system design motivations and validation, which is pre-requsites for a system paper like this one.

Main system design and methodological contribution is described in Section 3. With the current content, it is difficult to evaluate/comprehend the questions like, (1) what is the main methodological challenge to build the presented system? e.g., there is some description of having centralized and decentralized policy learning, etc, but whats the challenge and practical problems there? (2) what are design choices made, and why? e.g, the choice of a specific imitation learning method, specific way of obtaining the data, the choice of specific scenario for ware house, etc. I find it hard to get the main take-away from Section 3, apart from the description of how the authors designed the system without 'why' part.

- The paper's presentation needs more work.

Firstly, descriptions about the experimental design is hard to parse, despite being a crucial part to comprehend the overall empirical findings. How about showing in video material: the overall task description, what are the states (as in appendix), what the expert demonstrates, and what AR shows to the human user, and most importantly, qualitative comparison between each methods in the task executions? This would help to comprehend the material, and also have intuitive feeling on why VARIL can outperform other methods.

Second, introduction does not motivate human-multirobot set up, which is related to the impact of this work. Descriptions about the AR and the importance of adequate visualization is fine, but the considered scenario should also be given proper attention. This is because, the presented framework is shown to beat other baselines within this scenario, and it better be an important task in terms of application.

Third, methodological description in Section 3 requires more detailed descriptions. While presentation of the algorithm 1 helps in comprehensing the implementation details, more insights about the algorithm and descriptions of used methods incl. system design would be great to have in Section 3. Moreover, the paragraphs where the description jumped from line 1 to line 4 was difficult to understand. More materials like 'policy up' could be presented in the main paper, rather than in the appendix.

Lastly, there are also many missing notations in algorithm 1 which needs to be corrected in section 3. Many subsequent paragraphs and setnences start with 'We', which is a minor comment, but needs to be fixed.

- Ablation studies are missing, which can motivate step-by-step design choices of the designed system.

Why not try out other learning methods? Other classifiers other than SVM -- can it further boost the performance of the system? What is the specific use-case with respect to efficiency? e.g., when will the visualization policy be effective, when not?

- System paper only tested in simulation.

The authors do mention that having tested their system in simulation is one of their limitation, which will be hopefully addressed as future work. In my view, having a system paper only tested in simulation, is limited in its contribution and impact it can have to reach the bar for CoRL. When the system and ware house application in human-multirobot scenario is realized in physical mock-up, will the same scientific conclusion hold? Showing the impact of the system (incl. demonstration that the system can work in practice) is one of the missing steps in this paper.

**Summary Of Recommendation:**

I recommend weak reject. The paper itself presents an interesting system and experimental results in simulation demonstrates that it can outperform two other baselines.

However, this system paper lacks either thorough and novel methodological insights to design such a system, OR a practical impact, which is typically shown through hardware experiments that demonstrate the system's usage in real robotic applications.

----------------------------------------------- Update after rebuttal --------------------------------------------

I modified my recommendation to weak accept. The paper has its strength on developing interesting system, which is of relevance to robot learning researchers. The authors have also addressed my main issue on the lack of real robot experiments.

If the paper gets accepted, thoroughly checking the justifications of design choices and clarify of presentation in method section would be highly appreciated.

---

### Official Review · Reviewer_um9M · 2022-08-01

**Originality:** Very Good
**Technical Quality:** Very Good
**Clarity Of Presentation:** Excellent
**Impact:** 4

**Recommendation:**

Weak Accept: I recommend accepting the paper, but will not argue for my recommendation if the majority of other reviewers have a different opinion.

**Summary:**

This paper introduces a method for learning a policy to decide when to show which components of a visualization, in the context of Augmented Reality (AR) systems for human-robot collaboration. Existing literature provides baselines for sets of visualizations that are helpful during human-robot collaboration such as robot avatars, trajectories, etc. - which, according to the authors, can make for a busy AR interface in the presence of multiple robots and tasks. As a result, they propose VARIL which, using demonstrations from an expert, learns a policy to decide which components of the visualization are relevant at which points of the demonstration. They show that the learned visualization policies contribute to faster human-robot collaborations.

**Issues:**

After reading the experiment section I am still not sure exactly what the task was for the human worker (and have still not understood from playing for ~1min on the WebGL tool). It would be great if you could clarify this section.

Also, even if the details of how the DAgger-based algorithm need to be put in the appendix, I think it's important to include a rough idea of what the visualization policy state/action space looks like in the main paper. This is important because it's very hard to get a feel for the complexity of the visualization policy's task in the current state of the paper, which makes it harder to understand how important the human/policy help is.

**Quality Of The Limitations Section:**

Limitations are addressed clearly

**Reviewer Expertise:**

3: The reviewer is fairly confident that the evaluation is correct

**Robotics Focus:**

Highly relevant to robotics but no hardware experiments

**Strengths And Weaknesses:**

Strengths:

* The paper clearly introduces the motivation, literature, and establishes itself as both a new method and a new task (using learned policies for visualization).
* The experiments conducted seem to clearly indicate that learned policies, and the proposed learning method in particular, have a statistically significant effect on human-robot collaboration efficiency.

Weaknesses:

* The conducted experiment seems like it could be rather simplistic to understand the effects of such a visualization mechanism. Additionally, no experiments with real robots are performed. These create some doubt as to how the shown results will transfer to real-life use cases.
* It is not discussed why the authors believe imitation learning (especially in the form of DAgger) is the best paradigm for learning such a visualization policy. Indeed, in the presence of more switches, a human expert is likely to be strongly sub-optimal, and applying other paradigms such as RL on a simple MDP could provide above-human performance. The requirement of a human expert suggests that the provided visualization's value is upper-bounded by what a human expert engineer could heuristically program into the visualization software, beating the purpose of a learned policy.
* There are some missing components in the main paper that are shown in the appendix but could be included in the main paper (see issues).

**Summary Of Recommendation:**

I think that this paper presents an interesting new paradigm (learned visualization policies) and shows a limited, albeit convincing, demonstration that the idea is beneficial for the proposed goals. Indeed, while it might be possible to improve the policy learning process (especially in contexts where there are too many parameters for a human expert to toggle, or a human expert is not available), I believe that opening up this area of research is possibly very beneficial towards improving the state of AR for HRI and thus recommend that this paper be accepted.

---

### Official Review · Reviewer_XmHJ · 2022-08-02

**Originality:** Good
**Technical Quality:** Good
**Clarity Of Presentation:** Very Good
**Impact:** 3

**Recommendation:**

Weak Reject: I recommend rejecting the paper, but will not argue for my recommendation if the majority of other reviewers have a different opinion.

**Summary:**

====Summary: =====
This paper presents a method for learning to select visualizations (visualization agent) for on-line assistance in an augmented reality (AR) application for a human cooperating with a team of mobile robots in a warehouse-like scenario.  The visualization agent is trained through imitation learning (IL) using examples collected from a human expert providing labels for the desired visualization through a large number of episodes of the task with a human collaborating with the robot team. The method is tested through a user study that found the proposed method reduced task time when compared to other strategies for generating visualizations.


**Issues:**

====Comments: ====

*How much does the visualization policy depend on the specific motion planner and task allocation systems? Is the input from the expert generalizable beyond that?

*While using the on-line demo I noticed that some visualizations such as the future robot path is rendered on top of objects that hide the view to the robot. This is helpful to visualize the path, which would be otherwise occluded, but it’s also confusing as I didn;t fully understand where the trajectory was going towards. Did you do this in 3D virtual reality or on a computer screen?

*How does the sample complexity (number of demos from expert) scale with the size of the robot team? Is the learned policy expected to generalize in any aspect?

*As for the current version of the paper I don’t think it’s ready for conference publication because it lacks a full method explanation in the main paper. The paper needs to be self contained.

**Quality Of The Limitations Section:**

Limitations are addressed clearly

**Reviewer Expertise:**

4: The reviewer is confident but not absolutely certain that the evaluation is correct

**Robotics Focus:**

Highly relevant to robotics but no hardware experiments

**Strengths And Weaknesses:**

=====Strengths: =====

*The application of a learning method for creating a visualization agent that assists a human on-line during a collaborative task is creative and interesting. It’s a useful problem to study for robotics and human-robot interaction.

*The on-line demo of your unity system is very useful to understand the work, very nice! The paper is easy to read and clear, figures are clear and help understand the paper. In addition to the online demo, it would have been helpful to see actual episodes of users using the system for a task episode, including using the proposed method and the baselines.


====Weaknesses: ====

*I observed that the main paper seems to be limited in its explanation of the core learning method. It shows Algorithm 1 (VARIL), in which the policy update is done with PolicyUp. However, PolicyUp is only covered in supplementary materials. My main concern is that the paper is supposed to be self sufficient to explain the method, supplementary material should be left only for details, additional explanations, hyperparameter, etc, but not for materials core to understanding the contribution. It seems to me the learning part is imitation learning + Dagger, is this correct? Is there an innovation on the learning algorithm or how does it defer from other options algorithmically?

*In lieu of actual augmented reality (i.e. real environment, visualization overlap in glasses or similar device), the paper uses an implementation of this scenario in (what seems to be) virtual reality (a virtual scenario in Unity), and the visualizations are created in there. While this experimental method has a fair analogy with augmented reality and doesn’t evidently affect the data collection (expert recommended visualizations), keep in mind that might not be the same experience as actual augmented reality. As recognized in the Conclusions section, an implementation of this paper in augmented reality in the real world would require very advanced computer vision for detecting and tracking relevant objects, as well as accurate localization. A very challenging project that sounds very interesting, I don’t doubt applications like this will be fully developed.







**Summary Of Recommendation:**

*As for the current version of the paper I don’t think it’s ready for conference publication because it lacks a full method explanation in the main paper. The paper needs to be self contained.

---

### Meta-Review · Area_Chair_XL7L · 2022-08-13

**Recommendation:** Accept (Poster)
**Confidence:** 4

**Metareview:**

Summary:

In this paper, the authors proposed an AR space experimental field based on Unity to acquire human-robot collaboration strategies. The proposed methodology learns visualization policy through human demonstrations in the logistics domain of a warehouse. The results show an improvement in the efficiency of human-robot collaboration compared to the baseline.


Strength:

The online demo using the unity system is helpful for understanding the work.
The paper presents the system to learn visualization policy from human demonstrations.
Experimental results in simulation show that the proposed system can reduce the mental workload of the user and be more efficient when compared to existing approaches for visualization in AR.

Weakness:

The main policy update algorithm called PolicyUp is only covered in supplementary materials.
The authors used the keyword augmented reality; however, the experimental condition used in the evaluation is virtual reality. The gap between VR and AR should be carefully considered and explained in the paper.
As well as the lack of the actual AR condition, the real experiments using real robots is not performed.
It is not well discussed why imitation learning should be used in the framework. A more detailed discussion will be required to emphasize the strength of the proposed method.

Feedback:

Although some improvement was performed to the manuscript, it is hard to find some methodological lessons from the paper. There still are some rooms to improve the clarity. On the contrary, the paper shows an interesting application of robot learning techniques to human-robot interaction, which might be appreciated by certain researchers at CoRL. We hope the remaining questions could be solved in the revised paper and presentation.


**Best Paper Nomination:**

No